# Selective Encapsulation of the Polyphenols on Silk Fibroin Nanoparticles: Optimization Approaches

**DOI:** 10.3390/ijms24119327

**Published:** 2023-05-26

**Authors:** Oguz Bayraktar, Gizem Oder, Cansu Erdem, Merve Deniz Kose, Catalina N. Cheaburu-Yilmaz

**Affiliations:** 1Department of Bioengineering, Faculty of Engineering, Ege University, Bornova, 35100 Izmir, Turkey; gizemoderr@gmail.com (G.O.); cansuerdem3@gmail.com (C.E.); 2Department of Chemical Engineering, Faculty of Engineering, Ege University, Bornova, 35100 Izmir, Turkey; mervedenizkose@gmail.com; 3Laboratory of Physical Chemistry of Polymers, Petru Poni Institute of Macromolecular Chemistry, Romanian Academy, 41A Grigore Ghica Voda Alley, 700487 Iasi, Romania; 4Biochemistry Division, Department of Chemistry, Faculty of Science, Dokuz Eylul University, 35390 Izmir, Turkey

**Keywords:** silk fibroin colloidal solution, desolvation, selective encapsulation, nanoparticles, polyphenols, trans-resveratrol, quercetin, antioxidant

## Abstract

The present study proposes a method for designing small bioactive nanoparticles using silk fibroin as a carrier to deliver hydrophobic polyphenols. Quercetin and trans-resveratrol, widely distributed in vegetables and plants, are used here as model compounds with hydrophobic properties. Silk fibroin nanoparticles were prepared by desolvation method and using various concentrations of ethanol solutions. The optimization of the nanoparticle formation was achieved by applying Central Composite Design (CCD) and the response surface methodology (RSM). The effects of silk fibroin and ethanol solution concentrations together with the pH on the selective encapsulation of phenolic compounds from a mixture were reported. The obtained results showed that nanoparticles with an average particle size of 40 to 105 nm can be prepared. The optimized system for the selective encapsulation of the polyphenols on the silk fibroin substrate was determined to be 60% ethanol solution and 1 mg/mL silk fibroin concentration at neutral pH. The selective encapsulation of the polyphenols was achieved, with the best results being obtained in the case of resveratrol and quercetin and encapsulation of gallic and vanillic acids being rather poor. Thin-layer chromatography confirmed the selective encapsulation and the loaded silk fibroin nanoparticles exhibited antioxidant activity.

## 1. Introduction

There is a growing demand for phytomedicines and herbal extracts for various applications such as skincare, health care, functional food, nutraceuticals, and cosmetics, due to changes in consumer awareness of their health benefits [1]. Plant-based bioactive compounds have gained interest thanks to their antioxidant and antimicrobial properties [2,3] and their use within medical, cosmetic, food, and pharmaceutical industries has become competitive [4]. This increased demand was also induced by the misuse of antibiotics which has caused certain bacteria strains to develop resistance [5]. The biggest challenge concerning plant-based products is the isolation of one single bioactive compound from a mixture of bioactive constituents [6]. As extraction is the first and the most significant step for isolating plant-based bioactive phenolic compounds [7], many bioactive compounds are extracted as a mixture at the same time. In recent years, various methods were implemented for the isolation of the plant-based bioactive compounds [8]. However, there are limited studies dealing with the separation and purification of one single bioactive constituent from a mixture. After the separation of bioactive constituents, a second challenge, that of preserving bioactivity during processing conditions and storage, was reported. More specifically, the antioxidant activity of the polyphenols, being directly related to their bioactivity and stability, may suffer partial or total loss of activity during therapeutic usage.

Therefore, to maximize and preserve the bioactivity of natural compounds, encapsulation methods are preferred; the isolated bioactive compound can be protected from oxidation, dehydration reactions, and other unwanted reactions which may affect the bioactivity. Comunian et al. [9] reported a study where pomegranate seed oil (PSO) was encapsulated in conventional and Pickering emulsions using whey protein isolate (WPI) microgels (WPI natural and various combinations with gum arabic (GA), maltodextrin (MD), and modified starch (Capsul^®^). They protected PSO from its susceptibility to oxidation. The authors found out that a combination of whey protein isolates and modified starch were the best wall materials, resulting in the oxidative protection of the oil when compared with non-encapsulated oil.

Similarly, Tavares and Noreña [10] reported the encapsulation of ginger essential oil (GO) within whey protein isolate (WPI)/gum Arabic (GA) and GA/chitosan (CH) complex coacervates. The entrapment efficiencies of 55.31 and 81.98%, respectively, in the case of the complexes of GA/CH and WPI/GA, respectively, were found, revealing that GA/CH is a more efficient complex for GO protection. Bayraktar et al. [11] encapsulated trans-resveratrol in core/shell nanoparticles made of eggshell membrane proteins and silk fibroin, respectively, to preserve its stability by using coaxial electrospraying as the encapsulation technique.

An ideal encapsulation material should contribute to the functionality and stability of the active compound. Moreover, the selection criteria of the encapsulation material should include cost strains. Silk fibroin has been preferred in the literature due to its notable properties, such as biocompatibility and biodegradability, which are considered the most significant advantages over other synthetic and natural materials [12]. Silk fibroin was utilized in various applications, such as particles/nanoparticles [13,14], films [15], wound dressings [16], hydrogels [17], and scaffolds [18].

Calamak et al. [19] proposed silk fibroin as a material to be used for the development of new generation bionanotextiles such as wound dressings, bandages, tissue scaffolds. Silk-based bionanotextiles containing silver nanoparticles (AgNP) were manufactured via electrospinning, and process parameters were optimized. Methanol treatment and glutaraldehyde (GA) vapor seemed to influence the secondary structure of silk.

Zhang et al. [20]. prepared silk fibroin globular nanoparticles by using an aqueous silk fibroin solution regenerated with aprotonic and protonic water-miscible polar solvents. The obtained silk fibroin nanoparticles were not soluble but highly dispersed and stable in an aqueous solution. The resulting data showed that the degraded polypeptide components of these reconstructed silk fibroins were collected homogeneously or heterogeneously to form a looser spherical structure.

The desolvation method has been preferred in the literature to prepare nanoparticles due to its fast and straightforward fabrication of protein-based nanoparticles. It is a self-assembly process of the polymeric materials where a desolvating agent (e.g., alcohol or acetone) is added dropwise to an aqueous solution of protein under stirring to dehydrate the protein/natural polymer, resulting in conformational change from stretched to coil conformation. Various studies implemented the desolvation method for preparing albumin [21], serum bovine albumin [22], gelatin [23], and silk fibroin particles [24]. Many properties can be controlled via the desolvation technique such as controlled degradation, size, shape, and drug loading and release capacities of silk fibroin nanoparticles (SFNs). [25,26,27] Due to their small size, SFNs can go through thin capillaries, which improves the uptake of drugs by the targeted cells. Moreover, SFNs can be used for the targeted delivery of anticancer drugs to the tumor cells. Regarding this, silk fibroin nanoparticles have been used to encapsulate known anticancer drugs like doxorubicin [28], paclitaxel [29], and cisplatin [30].

The encapsulation of polyphenols can be achieved based on the interactions between protein and polyphenols. Proteins were used for the encapsulation of polyphenols to enhance their bioactivity and stability. Recently, enhancement of antioxidant capacity due to the interaction between rice protein and B-type procyanidin dimer (PB2) was reported by Dai et al. [31]. It was revealed that the presence of the PB2 decreased the α-helix and random coil structure of the rice protein and reduced its surface hydrophobicity because of the hydrophobic attractive forces forming molecular complexes between the protein and PB2.

Nanocarriers composed of whey protein isolate (WPI) were found to be suitable for encapsulating hydrophobic and amphiphilic bioactive compounds such as fatty acids, aromatic compounds, polyphenols, and vitamins to improve their solubility and chemical stability. Therefore, WPI was successfully used for the preparation of a nanocarrier for quercetin delivery [32].

A surface phenomenon including hydrogen bonding and hydrophobic interactions mainly governs the association between polyphenol and protein. The strength of this interaction is affected by being at or around the isoelectric point of the protein [33]. The structure of polyphenol and its hydroxylation degree determine the strength of the formed complexes depending on the existence of single or multiple hydrogen bonds within the complexes [34]. Apart from hydrogen bonds including polar groups, the interaction between proteins and polyphenols can occur because of the hydrophobic, non-polar aromatic rings of polyphenols and aromatic amino acids [35].

Whey protein nanofibril (WPF) was reported as an ideal natural nanoscale carrier for improving the water solubility of quercetin. The WPF and quercetin interact with each other through hydrophobic interactions and hydrogen bonding. It was shown that the stability of trans-resveratrol could be protected thanks to the interactions between polyphenols and proteins. Additionally, protein–polyphenol interaction contributed to the stability of the trans-resveratrol as well [11]. Polyphenols with a more hydrophobic character display a higher degree of protection, and with increasing hydrophilicity, protection becomes more difficult [36].

The present study aimed to prepare nanoparticles based on silk fibroin loaded with polyphenolic compounds’ mixture via a desolvation technique. The selective encapsulation of the four phenolic compounds (e.g., trans-resveratrol, quercetin, and gallic and vanillic acids) from a mixture was the expected outcome. The novel aspect of the present study is its contribution to the selective encapsulation approach. The study proposed first the detailed optimization of the system to avoid unnecessary and time-consuming experiments. Within this approach, the effect of silk fibroin and ethanol solution concentrations on the average particle size and selective encapsulation of phenolic compounds were investigated by using Design Expert software and theoretical experimental runs for the best optimization. By using the theoretical approach’s results, nanoparticles containing the phenolic mixture were prepared accordingly and analyzed in terms of structure and antioxidant activity.

## 2. Results and Discussion

### 2.1. Characterization of Silk Fibroin-Based Nanoparticles

Silk fibroin-based nanoparticles were characterized in terms of structure to identify the characteristics of both silk fibroin and polyphenol structure and possible interactions between the matrix and the bioactive compounds.

Silk fibroin, due to the presence of amide groups, showed characteristic vibration bands around 1620 cm^−1^; these were assigned to the absorption peak of the peptide backbone of amide I (C = O stretching), the bands around 1500 cm^−1^ to amide II (N-H bending), and the bands around 1200 and 1400 cm^−1^ to amide III (C-N stretching). Zhang et al. [37] determined the β-sheet conformation of silk fibroin presenting shifts of absorption peaks at 1620 cm^−1^ (for amide I), 1514 cm^−1^ (amide II), and 1230 cm^−1^ (amide III). They found out that the crystalline structure of CaCl_2_-ethanol silk fibroin showed more silk I (α-form, type II β-turn). Similarly, the silk fibroin used for the preparation of the nanoparticles had rather a β-sheet conformation mostly due to the fast-freeze-drying process presenting the characteristic vibration bands of amide at 1637 cm^−1^, 1516 cm^−1^, and 1237 cm^−1^.

As depicted in Figure 1, the characteristic vibration bands of the silk fibroin-based nanoparticles were assigned to the amide II groups (at 1665 cm^−1^ for NP and 1670 for NPPF), 1716 and 1719 for C = O stretching of NP and 1711 cm^−1^ for NPPF. C-N stretching was determined for the silk fibroin-based nanoparticles at 1400–1000 cm^−1^. Loading the polyphenols within the silk nanoparticle was evidenced by the presence of the C = O from 1700 cm^−1^ and esters and related moieties of C-O-C, C-O-O- within 1400–800 cm^−1^. When the polyphenols were loaded, the characteristic bands were observed qualitatively within the spectrum of the loaded nanoparticles (NPPF) together with some shifts of peaks indicating the H-bonds and electrostatic associations formed due to the loading and trapping inside the nanoparticles.

### 2.2. Characterization of the Phenolic Compound-Based Mixture

The mixture of the four polyphenols, i.e., quercetin, trans-resveratrol, vanillic acid, and gallic acid was analyzed in terms of its composition. The determination of the concentration of each polyphenol was assessed by comparing the theoretical values calculated with the experimental ones obtained by chromatographic separation. The calibration curves of gallic acid, vanillic acid, trans-resveratrol, and quercetin were done at 280, 257, 304, and 368 nm, respectively.

The obtained chromatograms are given in Figure 2 and the information on the content of the polyphenols is listed in Table 1.

Following the chromatograms, the 3D chromatogram of the phenolic compounds described the polyphenol mix (PFmix) 3D distribution (Appendix A).

As seen in Figure 3, four polyphenol compounds representing a range of different hydrophobicities were chosen in the phenolic compound mixture. The hydrophobicity of the polyphenols estimated from the partitioning between octanol and water were given as LogP values (ChemAxon).

The objective of this study was to investigate the effects of the presence of polyphenols displaying a varying degree of hydrophobicity on the size of silk fibroin nanoparticles and its selective encapsulation performance. For the flavonoids, LogP values decreased with the increase in the number of hydroxyl groups in the structure, but the addition of a galloyl group can inversely affect this value.

### 2.3. Effect of Silk Fibroin and Ethanol Concentration on Average Particle Size

A total of 12 experiments were performed to evaluate the effects of silk fibroin and ethanol concentrations on average particle size. The experimental CCD and observed responses are given in Table 2.

Based on the R^2^ values obtained, the model was best fitted to the reduced quartic model. For the fitted model, the obtained analysis of variance values is given in Table 3.

As seen in Table 3, the *p*-value was less than 0.05, which indicates that model terms are significant. When F values were compared, it was concluded that ethanol concentration has a higher impact on the average particle size than silk fibroin concentration. In this case, A, B, AB, A², B², A²B, A²B² are significant model terms. According to the results of the statistical analysis, Equation (1) was obtained for average particle size. Equation (1) could be used to make predictions about the average particle size for given levels of each factor.
(1)$$Particle\;Size\left(nm\right)=58.38+36.46*A-48.96*B-26.25*AB+\;33.24{*A}^{2}+43.66*B^{2}+102.71*A^{2}B+14.79*AB^{2}+85.97A^{2}B^{2}$$

In Figure 4, the 3D response surface graph and statistical approval of the reduced quartic model are given.

From Figure 4a, it can be observed that both silk fibroin and ethanol solution concentration affected the average particle size. For instance, at 60% EtOH concentration, when silk fibroin concentration increased, the average particle size was changed from 50 to 250 nm. At a relatively higher EtOH concentration of 80%, when silk fibroin concentration increased from 1 to 1.5 mg/mL, the average particle size first decreased from 250 to 100 nm, then increased to 300 nm with the increase of silk fibroin concentration to 2 mg/mL. At a lower silk fibroin concentration of 1 mg/mL, average particle size increased from 60 to 250 nm with the increase of EtOH concentration from 60 to 80%. On the other hand, at relatively higher silk fibroin concentrations (2 mg/mL), the increase in EtOH concentration from 60 to 70% caused average particle size to decrease from 250 to 100 nm; then, an increase in EtOH concentration to 80 % resulted in particles with an average size of 300 nm. The minimum average particle size can be obtained at the center point of the experimental design. From Figure 4b, it can be understood that a good fit between predicted values and the experimental data points occurred. In Figure 5, the desirability ramp for optimization for the minimum particle size is given.

Desirability ramp optimization showed that to fabricate the particles with the minimum average particle size of 44 nm, concentrations of silk fibroin and ethanol solution should be chosen as 1.44 mg/mL and 76%, respectively. Fabrication of SFNs was done experimentally at given process parameters. In Figure 6a,b, the histogram graph (a) and AFM image (b) of the prepared particles with an average particle size of 40 nm are given.

Figure 6a shows that silk fibroin particles prepared with the process parameters obtained from the desirability ramp-predicted value were achieved experimentally. These results also validated the accuracy of the obtained model.

### 2.4. Effect of Silk Fibroin and Ethanol Concentration on Encapsulated Phenolic Compound Concentration

A total of 12 experiments were performed to evaluate the effects of silk fibroin and ethanol concentrations on the average particle size and each phenolic compound’s concentration. The experimental CCD and observed responses are given in Table 4.

As seen in Table 4, with phenolic compounds’ addition to the desolvation process, the minimum average particle size for the obtained nanoparticles increased from 50 to 150 nm. This increasement tendency of the particle size might be due to the existing interaction between phenolic compounds ‘molecules and silk fibroin chains. As seen in Table 5, gallic acid and vanillic acid encapsulation from the phenol mix could not be achieved effectively, in contrast to the trans-resveratrol and quercetin. This outcome could be explained by the different solubility of vanillic acid and gallic acid in ethanol in comparison with aqueous environments. Details on the polyphenolic compounds from the polyphenolic mixture (PFmix) concerning molecular weight and calculated logP (ChemAxon) are summarized in Table 3. Gallic acid, vanillic acid, quercetin, and trans-resveratrol have LogP values of 0.72, 1.17, 2.16, and 3.40, respectively (Table 3). Quercetin is a well-known hydrophobic compound that has solubility in an aqueous solution ranging from 0.00215 g/L to 25 °C to 0.665 g/L at 140 °C [38]. Quercetin has a high solubility in organic solvents such as ethanol (approximately 2 g/L) with a LogP value of 2.16. Vanillic acid has a moderately lipophilic octanol–water partition coefficient (Pow between 10 and 100, which corresponds to a logP between 1 and 2) of 1.17. Unfortunately, compounds with a logP value between 1 and 2 are mostly sparingly soluble either in water or in oil. Trans-resveratrol, which is the biologically active isomer of resveratrol, is a very hydrophobic molecule, practically insoluble in water (water solubility being 0.069 g/L), and relatively neutral. The solubility of resveratrol in alcohol is probably due to the hydrophobicity of resveratrol (logP = 3.40). As expected, hydrophobicity is dependent on the substitution of the hydroxyl group by a glucose residue, which makes quercetin less hydrophobic than resveratrol. However, both are quite hydrophobic molecules compared with gallic and vanillic acid.

With the addition of silk fibroin solution to the ethanol solution, hydrophobic interactions increased, which explained the encapsulation of the trans-resveratrol and quercetin to the silk fibroin nanoparticles due to their low solubility in water [39].

Silk fibroin is an amphiphilic polymer whose large hydrophobic domains occupy most of the polymer, which has a high molecular weight. Hydrophobic areas are interrupted by small hydrophilic spacers, and the N- and C- terminus of the chains are also very hydrophilic. The hydrophilic-hydrophobic environment in the presence of molecules affects the structural state of silk fibroin and its interaction with the molecules around [40]. Compounds can be either encapsulated in their central core or bound to the surfaces by electrostatic or hydrophobic interactions with silk fibroin.

The highest encapsulation of phenolic compound was achieved with 60% ethanol solution and 1 mg/mL silk fibroin concentration, and 59% of the encapsulated phenols were trans-resveratrol. The data given in Table 5 were examined with the analysis of variance, and the obtained results are given in Table 5.

Generally, the larger the magnitude of the F-value and the smaller the *p*-value, the more significant the effect of the corresponding parameter on the response [41]. As can be observed in Table 5, based on the F-values, the most effective parameter was found to be ethanol concentration for quercetin concentration and average particle size, while for trans-resveratrol, the most effective parameter was silk fibroin concentration.

In Figure 7, the 3D response surface graphs of each response for the reduced quartic model are given. The interactive effects of the parameters are shown in Figure 6. As seen in Figure 6, the red regions show the highest response values, and the yellow and blue parts represent the lower and much lower values.

Figure 7a shows that at 80% ethanol concentration, when silk fibroin concentration decreased to 1 mg/mL, the average particle size increased from 50 nm to 150 nm. As seen in Figure 7b, at 2 mg/mL silk fibroin concentration, when ethanol concentration decreased from 80% to 60%, the encapsulated gallic acid amount increased by 0.28 μg. Figure 7c shows that silk fibroin concentration was 1 mg/mL when ethanol concentration changed from 80% to 60%; no vanillic acid was encapsulated. However, at 60% ethanol concentration, when silk fibroin concentration increased from 1 mg/mL to 2 mg/mL, 2 μg vanillic acid was encapsulated. When silk fibroin concentration changed from 1 mg/mL to 2 mg/mL at 80% ethanol solution, the encapsulated trans-resveratrol amount was increased from 1 μg to 1.8 μg. As seen in Figure 7e, when ethanol concentration changed from 80% to 60% at 1 mg/mL silk fibroin solution, the encapsulated quercetin amount was doubled from 1 μg to 2 μg.

Based on the experimental design, minimum average particle size and maximum encapsulated trans-resveratrol values can be achieved with 60% ethanol solution and 1 mg/mL silk fibroin concentration. In Figure 8, the desirability ramp for optimization for the minimum particle size is given. The prepared silk fibroin particles with a minimum average particle size of 183 nm could encapsulate hydrophobic compounds, quercetin, and trans-resveratrol.

The proposed explanation is based on the ability of some groups to determine the presence of residues of hydrophilic amino acids. This makes it possible to obtain polyphenols with high hydroxyl groups not only to establish hydrophobic interactions with the hydrophobic part of the protein but also to increase the interaction energy thanks to the hydrogen bond [42]. The same mechanism may likely explain why silk fibroin at relatively higher concentrations may achieve significantly higher selectivity for the selective encapsulation of trans-resveratrol/quercetin. A mixed-mode behavior may be available, based upon both hydrophobic and hydrogen-bond interactions. This may add a further degree of freedom, which may affect both the geometry and the strength of the adsorption interaction and therefore increase selectivity [42].

The selective encapsulation of polyphenols from a mixture in the presence of the silk fibroin nanostructures (nanoparticles) increased with increasing partitioning of a polyphenol towards the hydrophobic environment.

### 2.5. Effect of pH on the on Encapsulated Phenolic Compound Concentration

To determine the effect of pH on silk fibroin nanoparticles’ formation and encapsulation of phenolic compounds, preliminary studies were conducted. Preliminary studies showed that silk fibroin nanoparticles were not obtained in the alkaline pH. The supernatant was analyzed by the HPLC method. The results showed some degradation of the phenolic compounds due to pH. Resveratrol, a phenol-like compound, becomes ionized when the pH increases in the basic medium and causes the compound to be unstable, resulting in rapid degradation.

Friedman and Jürgens studied the effect of pH on the stability of plant phenolic compounds. The results showed that caffeic, chlorogenic, and gallic acids are not stable to high pH and that the pH and time-dependent spectral transformations are not reversible [43]. Due to this reason, in this study, pH values between 2 to 7 were chosen to be studied.

A total of 18 experiments were performed to evaluate the effects of silk fibroin, ethanol concentrations, and pH on the average particle size and each phenolic compound’s concentration. The experimental CCD and observed responses are given in Table 6.

As seen in Table 7, the obtained data matched with the data given in Table 6. The pH change was not effective on the phenolic compounds’ selective encapsulation; however, a slight increment in the average particle size was observed.

The optimization for the minimum average particle size and maximum encapsulated trans-resveratrol desirability ramp in Figure 7 was done in Std 5. The stated results were obtained as particle size 120 nm and encapsulated trans-resveratrol 0.0024 mg, which is 62% of the encapsulated phenolic compounds.

The data given in Table 7 were examined with the analysis of variance, and the obtained results are given below.

In Figure 9, the 3D response surface graphs of each response for the reduced quartic model are given. The interactive effects of the parameters are shown in Figure 9. The red regions show the highest response values, and the yellow and blue parts represent domains with low response values.

Based on the performed optimization by experimental design, the minimum average particle size and maximum encapsulated trans-resveratrol values were established to be 60% ethanol solution and 1 mg/mL silk fibroin concentration. In Figure 10, the desirability ramp for optimization for the minimum particle size is given.

The values of the parameters for desirability ramps from Figure 7 and Figure 9 were similar, showing that pH within the range of 2 to 7 was not an effective parameter for the selective encapsulation of phenolic compounds and the obtained experimental design models were correct. In addition, by changing the process parameters, it is possible to encapsulate the desired compound.

### 2.6. Determination of Effect of Phenolic Compounds on the Selective Encapsulation

In order to determine the effect of phenolic compounds present in the phenol mix on the selectivity of encapsulation, they were checked with the same amount of phenolic compounds separately. Experiments were done with 2 mg/mL silk fibroin solution and 80% ethanol solution. The obtained data are given in Table 8.

Table 8 shows that hydrophobic interactions were controlled by the encapsulation of phenolic compounds. When the compounds were added separately to the desolvation process, the same results were obtained as for the phenol mix. This result suggested that phenolic compounds’ interaction in the desolvation process does not change the phenolic compounds’ encapsulation observed in the mixture of phenolic compounds. The selective encapsulation changed with silk fibroin and ethanol solution concentrations.

When the results were analyzed, it was seen that when the ethanol concentration changed from 60% to 80%, the most significant change occurred on the encapsulated trans-resveratrol with 1 mg/mL silk fibroin concentration.

At 80% ethanol solution, when silk fibroin concentration increased from 1 mg/mL to 2 mg/mL, it was seen that encapsulated trans-resveratrol and the gallic acid amount were increased. The most significant encapsulated gallic acid and vanillic acid amounts were obtained at 60% ethanol solution.

### 2.7. Thin-Layer Chromatography (TLC) and Antioxidant Activity of the Selective Encapsulated Polyphenols

TLC chemical screening coupled with image analysis detection was evaluated for the quantitative determination of studied polyphenols [44,45]. According to Figure 11, the recorded images allowed a visual evaluation of the presence of the polyphenolic acids within silk nanoparticles. The method was suitable for rapid quantification of the four polyphenols from the mixture and it allows a rapid quantification of the loading efficiency.

Figure 11A represents the TLC screening of the polyphenolic mixture (PF mix) and the pure polyphenols, their solutions being prepared in the same conditions as for PFmix. The obtained outcome of the solubility differences and hidrophobicity was supported by the TLC analysis, as it was observed that the lowest Rf value was obtained in the case of gallic acid migrating less. The migration of PFmix loaded within the silk fibroin nanoparticles is shown within the Figure 11B series and the determined values for Rf were listed in Table 9.

By comparing the Rf values of the polyphenols and analyzing the migration band of PFmix of the encapsulated polyphenols, the values are comparable, and the positions of the migration bands are similar with the pure constituents leading to the confirmed outcome that the polyphenols were successfully encapsulated within the silk fibroin nanoparticles. The antioxidant capacity of encapsulated mix polyphenols (NPPF) and encapsulated polyphenols (NPQ, NPR, NPGa and NPVa) are represented within Figure 12. The highest calculated activity was determined for NPPF where the synergetic activity of the encapsulated polyphenols was observed. The biggest contribution to this antioxidant activity seemed to be the presence of quercetin and resveratrol, as their affinity to be encapsulated faster and selectively was determined in the previous sections.

When analyzing the antioxidant activity of the standards alone, the values of the index describing Trolox inhibition (as mg/g) are much higher compared with the values obtained for the encapsulated polyphenols. However, this comparison cannot be fully assessed as the both the encapsulation process and combination of the four polyphenols affected the antioxidant activity. Interestingly, the inhibitory activities of quercetin and trans-resveratrol are lower when acting alone but within the nanoparticles they showed higher activity when compared with the nanoparticles of gallic and vanillic acid. The results obtained regarding the antioxidant activity of trans-resveratrol and quercetin were comparable with the ones found by other researchers as well [11,46,47].

## 3. Materials and Methods

### 3.1. Materials

HPLC-grade acetonitrile and methanol were purchased from Carlo Erba, Spain. All chemicals were of analytical grade. Quercetin and vanillic acid were purchased from Fluka, Honeywell, Charlotte, North Carolina, USA. Sodium carbonate (anhydrous) and calcium chloride (anhydrous) were purchased from Merck, Darmstadt, Germany. Trans-resveratrol and gallic acid were purchased from Sigma Aldrich, Darmstadt, Germany. Dialysis tube (SnakeSkin TM Dialysis Tubing, 10K MWCO, 22 mm) was purchased from Thermo Fischer, Waltham, Massachusetts, USA. Raw Bombyx mori silk was obtained from Bursa Institute for Silkworm Research (Bursa, Turkey). ABTS assays were performed by using ABTS (2,2′-Azinobis-(3-Ethylbenzthiazolin-6-Sulfonic Acid)), purchased from Roche (Mannheim, Germany) and potassium persulfate (K_2_S_2_O_8_) from Carlo Erba Reagents (Val-de-Reuil, France). Trolox standard was achieved from Acros Organics (Geel (Belgium). Methanol, a Merk product (99% purity), was used as such, for dilution and solubility experiments.

### 3.2. Preparation Methods

#### 3.2.1. Preparation of Silk Fibroin Solution

The sericin was removed from raw silk with aqueous 0.5 % (m/v) Na_2_CO_3_ solution by boiling three times for 30 min each time. Degummed silk was washed with distilled water then dried at room temperature. To obtain a silk fibroin solution, Ajisawa solution was preferred (CaCl_2_: ethanol: water (111: 92: 144)) [48]. The solution and silk fibroin were mixed at 70 °C for 2 h. Then, the prepared solution was dialyzed with a dialysis membrane (SnakeSkin TM Dialysis Tubing, 10K MWCO, 22 mm) with distilled water at 4 °C for 3 days to obtain a silk fibroin solution.

#### 3.2.2. Preparation of Phenolic Compounds Mix

To obtain the same amount of molecule for each phenolic compound in the phenol mix, each solution was prepared with 4.118 × 10^−6^ mol. As a result, a concentration of each compound’s stock solution was prepared as Ctres = 0.94, Cquer = 1.246, CVA = 0.69, CGA = 0.70 mg/mL, respectively. The phenolic compound mix was prepared as 3 mL of each phenolic compound solution mixed with 24 mL 70% ethanol solution for better compliance with desolvation studies.

#### 3.2.3. Preparation of Silk Fibroin Nanoparticles

The desolvation technique was preferred for the preparation of silk protein fibroin nanoparticles using various concentrations of ethanol solutions. To induce the desolvation process, 8.5 mL of ethanol solution was placed into a small beaker, and 1.5 mL of various concentrations of silk fibroin solution (Table 2) was added dropwise at room temperature with constant stirring at a very low rpm for 2 h. After desolvation, silk fibroin nanoparticles were centrifuged twice at 6000 rpm for 20 min to obtain the precipitates. The supernatant was removed, and silk fibroin precipitates washed with distilled water.

#### 3.2.4. Preparation of Silk Fibroin Nanoparticles Loaded with Bioactive Compounds

To induce the desolvation process, 7.5 mL ethanol solution was mixed with a 1 mL phenolic compound mix. Then, a 1.5 mL silk fibroin solution was added dropwise at room temperature with constant stirring at a very low rpm for 2 h. After desolvation, silk fibroin nanoparticles were centrifuged twice at 6000 rpm for 20 min to obtain the precipitates. The supernatant was removed, and silk fibroin precipitates were washed with distilled water. Selective encapsulation of four polyphenols representing a range of different hydrophobicities including gallic acid, vanillic acid, trans-resveratrol, and quercetin in aqueous ethanolic solutions was monitored by HPLC.

### 3.3. Characterization Methods

#### 3.3.1. Fourier Transform Infrared Spectroscopy (FTIR) Analysis

The structure was confirmed by FT-IR spectra of the copolymer films which were recorded with a Perkin-Elmer Spectrum-100 ATR-FTIR instrument by scanning in the range of 600–4000 cm^−1^.

#### 3.3.2. HPLC Analysis

The chromatographic separation was done by a modified method of Delgado et al. [39]. The chromatographic separation was carried out using isocratic mobile phase methanol–acetic acid-water (52:2:46) as a solvent with a flow rate of 1.0 mL/min. The HPLC column was equipped with a C18-type reversed-phase column (5 μm, 4.6 × 250 mm, Acclaim™ 120, Thermo Fischer Scientific, USA) and the analyses were carried out by using an UltiMate 3000 Chromatograph (Thermo Scientific, USA). The dried samples were dissolved in methanol–water (1:1), filtered and aliquots of 25 µL were injected into the HPLC system. The detection of trans-resveratrol, quercetin, vanillic acid, and gallic acid was established at 280, 257, 304, and 368 nm, respectively. Triplicate injections were performed, and average peak areas emission wavelengths were used for the quantitation. For each compound, readings were done at their maximum absorbance wavelength.

#### 3.3.3. Experimental Design

Based on our preliminary studies and the literature, the highest silk fibroin concentration was chosen as 2 mg/mL. Effects of silk fibroin concentration and ethanol solution concentration were investigated using Central Composite Design (CCD) in response surface methodology (RSM) by Design Expert Version 12.0.0 (Stat-Ease Inc., Minneapolis, MN, USA). The average particle size was chosen as the response for each experimental setup. Independent variables and their levels used for CCD are tabulated in Table 10. Center point (silk fibroin concentration: 1.5 mg/mL, ethanol concentration: 70% (*v/v*)) was repeated four times in the prepared design. Two experimental set-ups were prepared with and without phenolic compounds under the same conditions. In experimental setup 2, which worked with phenolic compounds, in addition to the average particle size, the concentration of trans-resveratrol, gallic acid, quercetin, and vanillic acid were chosen as responses. Moreover, the effect of pH was investigated with CCD within experimental set up 3.

#### 3.3.4. Particle Size Analysis

The average particle size was determined with an atomic force microscope (AFM) (ezAFM, Nanomagnetics Instruments, Ankara, Turkey) was used. A histogram of the particles was obtained with AFM with tapping mode.

#### 3.3.5. Thin-Layer Chromatographic Chemical Screening (TLC)

Thin-layer chromatography (TLC) is a method used to separate components in a mixture. With this method, the purity of the mixtures can be determined. After separation, each compound shows horizontal bands at different points. Each band has a specific retention factor (Rf) which can be determined by using Equation (2):(2)$$Rf=\frac{distance\;traveled\;by\;sample}{distance\;traveled\;by\;solvent}$$

The encapsulated phenols were dissolved in ethanol with a concentration of 1 mg/mL. Chloroform: methanol (9:1) solvent mixture used as mobile phase to screen and determine the separation. Approximately 10 mL of mobile phase was placed into a rectangular glass tank and covered with a glass lid; the solvent-based mixtures were allowed to saturate for 10 min before use. Then, 2 μL of each sample were placed with a capillary tube on the DC Kieselgel 60 F₂₅₄ silica gel (0.2-mm thickness were purchased from Merck), totalling 5 drops. Unloaded phenolic solutions were used as a reference sample and placed onto silica gel as well. After separation, spots were detected at 254 nm.

#### 3.3.6. ABTS Assay

ABTS assays were performed to determine the antioxidant activity of the prepared nanoparticles by comparison with the pure constituents [49]. The aim of the assay was also to obtain information on the capacity of preservation of the bioactive characteristics of the encapsulated polyphenols from the mixture. ABTS●+ radical cation was produced by reacting 7 mM ABTS and 2.45 mM potassium persulfate (K_2_S_2_O_8_) at a ratio of 1:1 by incubation in the dark at room temperature for 16 h. After 16 h, ABTS solution was diluted with methanol in the spectrophotometer until it reached absorbance values of 0.7 at 734 nm. Trolox was used as a reference standard. Silk fibroin samples were prepared at a concentration of 0.5 mg/mL and 10 μL of sample or standard was added to the wells of 96-well plates in 3 replicates. Then, 200 µL ABTS solution was added to each well, mixed, and incubated in the dark at room temperature for 30 min. Absorbance measurements were taken at 734 nm using a microplate reader. By using the same conditions, the four polyphenols were analyzed as standards, by using a concentration of 0.025 mg/mL. Results were expressed as Trolox equivalent antioxidant capacity (TEAC). Trolox equivalent antioxidant capacity refers to the Trolox concentration with the same antioxidant capacity as the sample. Antioxidant capacity is defined as µg equivalents of Trolox solution per gram of sample (µmol Trolox equivalent/g sample).

## 4. Conclusions

The present study proposes a method for designing a small bioactive nanoparticle using silk fibroin as a carrier to deliver hydrophobic polyphenols. Quercetin and trans-resveratrol widely distributed in vegetables and plants are used here as model compounds with hydrophobic properties. Hydrophobic attractive forces between polyphenols and silk fibroin have hydrophobic properties in terms of capacity and selectivity toward trans-resveratrol, and quercetin could be an effective way of purifying polyphenols from complex mixtures. The study’s approach is to first optimize polyphenols’ separation from a mixture by using Design Expert, thus avoiding unnecessary experiments, and loss of time and materials. The optimization of the nanoparticle formation by desolvation method was performed by using Central Composite Design (CCD) and the response surface methodology (RSM). The optimization methodology provided accurate and reliable data, revealing that it is possible to encapsulate phenolic compounds selectively by changing the desolvation process parameters; the model fitted the experimental data. Desirability ramp optimization showed that, to fabricate the particles with minimum average particle size and maximum encapsulation of trans-resveratrol, concentrations of silk fibroin and ethanol solution should be chosen as 1 mg/mL and 60%, respectively. Thin-layer chromatography confirmed that the selective encapsulation and the loaded silk fibroin nanoparticles exhibited antioxidant activity, proving its usefulness in natural-based end products such as nutraceuticals, cosmetics, or drug delivery systems where antioxidant activity is needed.

## Figures and Tables

**Figure 1 ijms-24-09327-f001:**
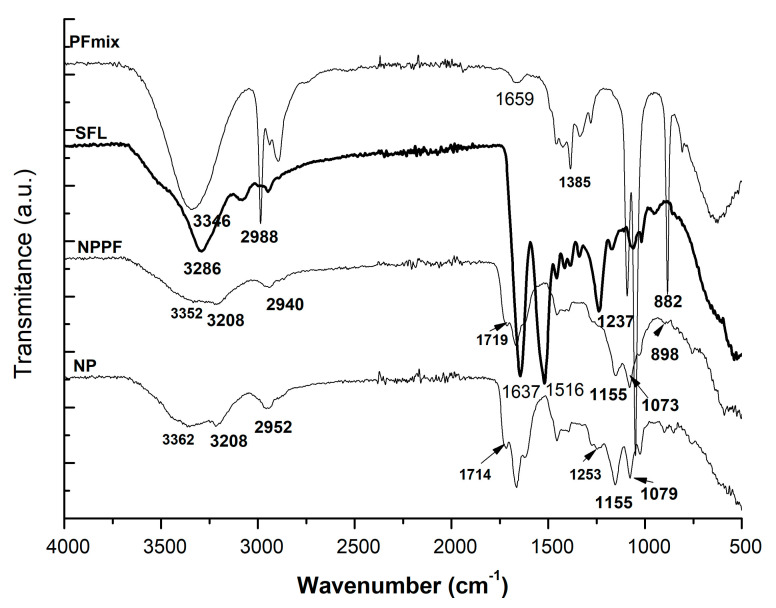
FTİR spectra of nanoparticles of silk fibroin loaded and unloaded with polyphenol mixture: SFL—silk fibroin lyophilized; NP—unloaded nanoparticles based of silk fibroin; NPPF—polyphenols loaded silk fibroin nanoparticles; PFmix—polyphenolic mixture.

**Figure 2 ijms-24-09327-f002:**
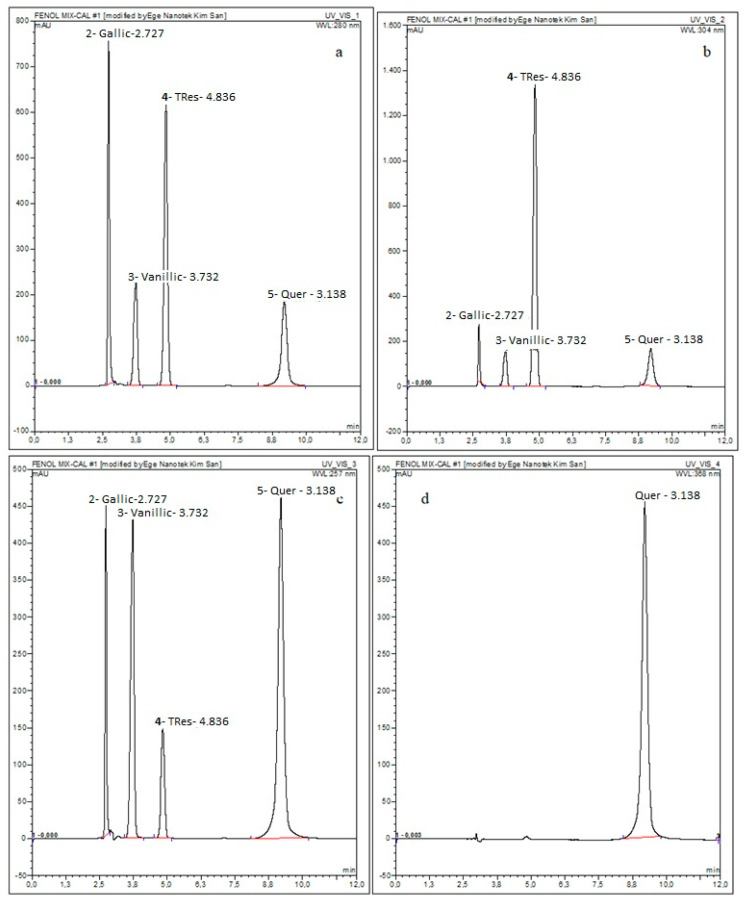
HPLC Chromatograms of the prepared phenolic compound mix at (**a**) 280 nm (Ga); (**b**) 304 nm (Res); (**c**) 257 nm (Va); and (**d**) 368 nm (Q).

**Figure 3 ijms-24-09327-f003:**
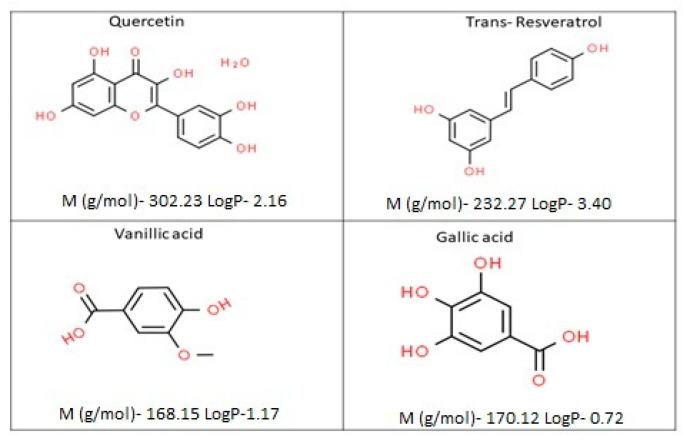
Schematic representation of the molecular structures of the studied polyphenols together with their structural characteristics.

**Figure 4 ijms-24-09327-f004:**
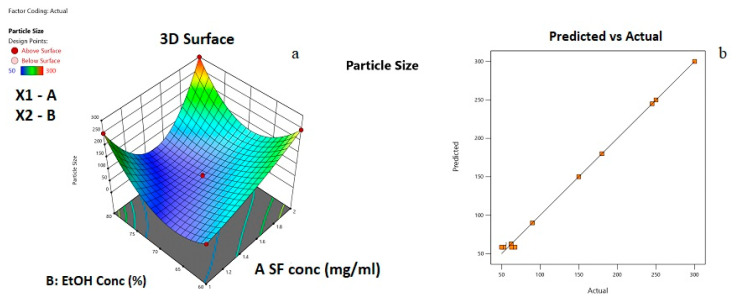
(**a**) Response surface plot for the effects of independent variables on average particle size; (**b**) Plot of the experimental and predicted values of responses for average particle size.

**Figure 5 ijms-24-09327-f005:**
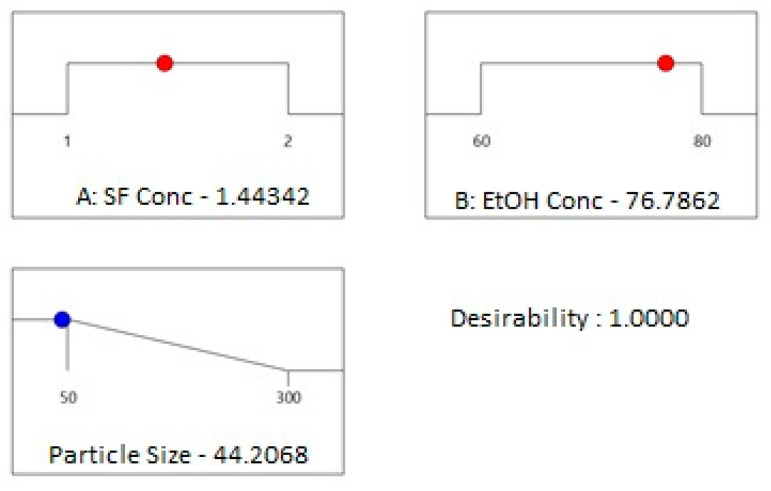
Desirability ramp for optimization for the minimum average particle size.

**Figure 6 ijms-24-09327-f006:**
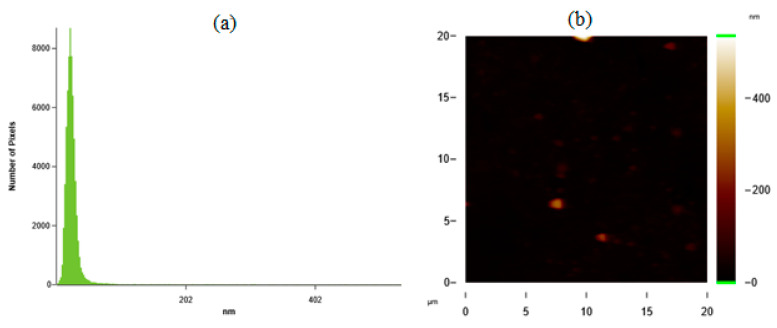
(**a**) Histogram graph; (**b**) AFM image of the prepared particles for the minimum average particle size.

**Figure 7 ijms-24-09327-f007:**
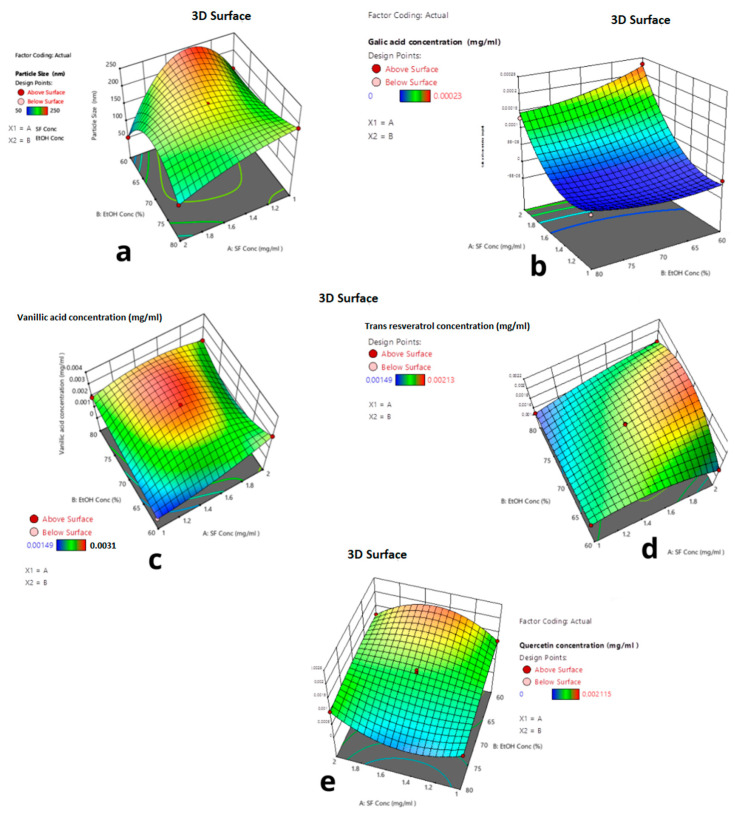
Three-dimensional graphs of responses depending on ethanol concentration and silk fibroin concentration. (**a**) Particle size (**b**) Gallic acid (**c**) Vanillic acid (**d**) Trans-resveratrols. (**e**) Quercetin concentration.

**Figure 8 ijms-24-09327-f008:**
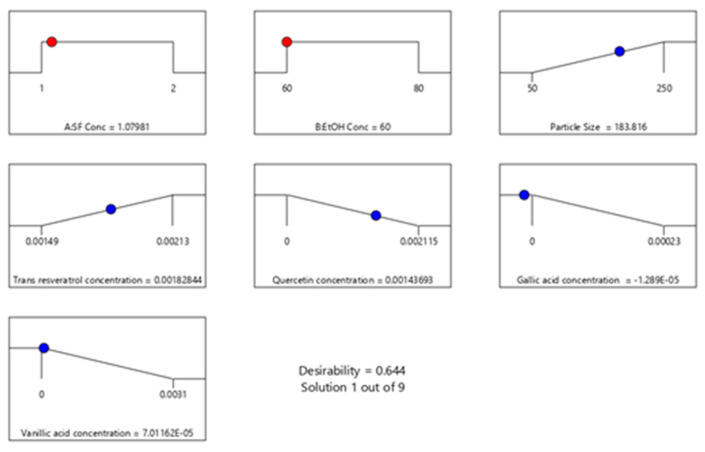
Desirability ramp for optimization for the minimum average particle size and maximum encapsulated polyphenols.

**Figure 9 ijms-24-09327-f009:**
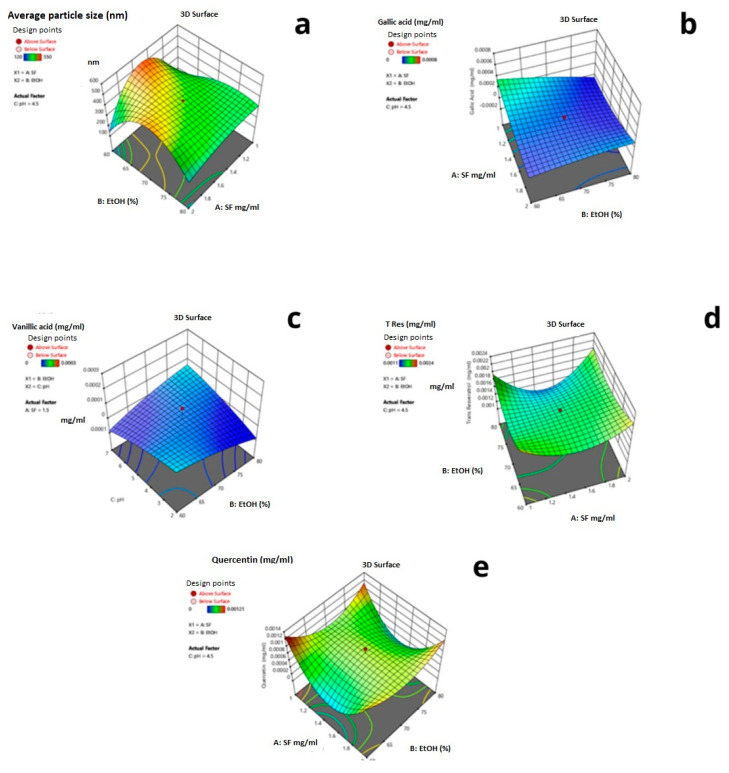
Three-dimensional graphs for the maximum and minimum values of the response variables: (**a**) Particle size (**b**) Gallic acid (**c**) Vanillic acid (**d**) Trans-resveratrol (**e**) Quercetin concentrations.

**Figure 10 ijms-24-09327-f010:**
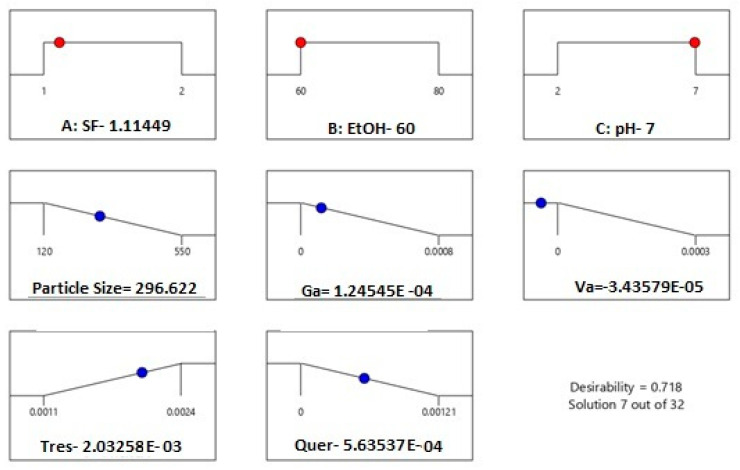
Desirability ramp for optimization for the minimum average particle size and maximum encapsulated trans-resveratrol.

**Figure 11 ijms-24-09327-f011:**
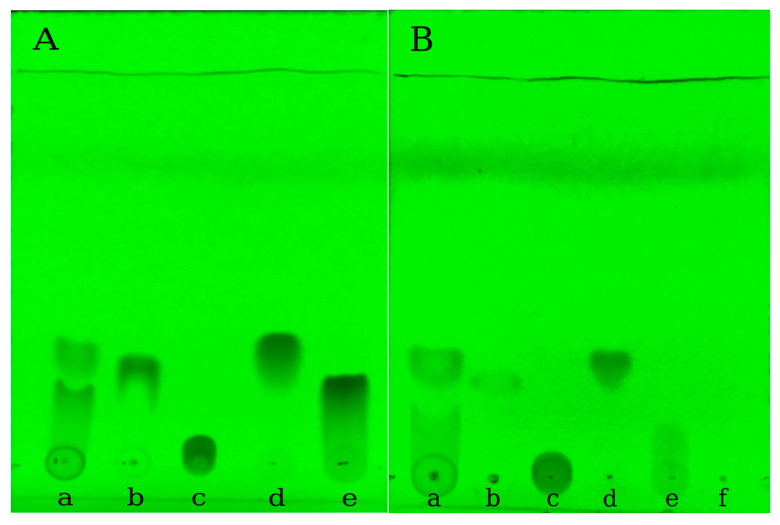
The chromatographic separation of phenolic mixture. (**a**) Trans-resveratrol; (**b**) Gallic acid; (**c**) Vanillic acid; (**d**) Quercetin; (**e**) Control; (**f**) UV images at 254 nm of (**A**) unloaded samples and (**B**) encapsulated samples.

**Figure 12 ijms-24-09327-f012:**
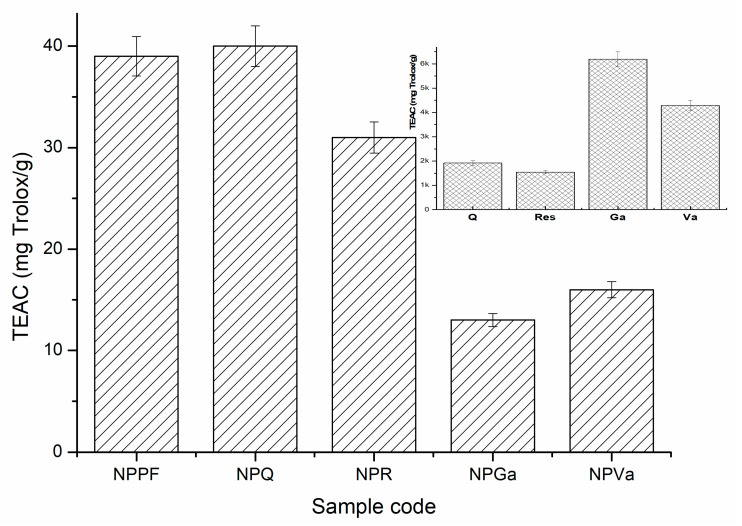
Trolox equivalent antioxidant capacity (TEAC) of encapsulated mix polyphenols (NPPF) and encapsulated polyphenols (NPQ, NPR, NPGA, and NPVa).

**Table 1 ijms-24-09327-t001:** Theoretical and Experimental Concentrations of Each Phenolic Compound from the Phenol Mix.

	The Concentration of Phenolic Compounds (mg/mL)			
	Gallic Acid	Vanillic Acid	Trans-Resveratrol	Quercetin
Experimental (HPLC)	0.0548	0.0430	0.0768	0.1010
Theoretical	0.0583	0.0500	0.0783	0.1030

**Table 2 ijms-24-09327-t002:** Central Composite Design (CCD) matrix and experimentally observed responses in experimental set up 1.

Exp No	Independent Variables		Observed Response
	SF Concentration (mg/mL)	EtOH Concentration (%)	Average Particle Size (nm)
1	0.9	70	62.5
2	2	60	245
3	1.5	70	50
4	1.5	70	67.5
5	1.5	82	62.5
6	1	80	250
7	1.5	58	180
8	1	60	90
9	1.5	70	63
10	2	80	300
11	2.1	70	150
12	1.5	70	53

**Table 3 ijms-24-09327-t003:** Analysis of variance (ANOVA) for reduced quartic function model for average particle size.

Source	Sum of Squares	df	Mean Square	F-Value	*p*-Value
Model	90,639.97	8	11,330.00	166.72	0.0007
A-SF Concentration	3828.13	1	3828.13	56.33	0.0049
B-EtOH Concentration	6903.13	1	6903.13	101.58	0.0021
AB	2756.25	1	2756.25	40.56	0.0078
A²	3055.38	1	3055.38	44.96	0.0068
B²	5270.18	1	5270.18	77.55	0.0031
A²B	17,663.44	1	17,663.44	259.92	0.0005
AB²	366.35	1	366.35	5.39	0.1029
A²B²	9597.11	1	9597.11	141.22	0.0013
Pure Error	203.87	3	67.96		
Cor Total	90,843.84	11			

Model: Significant, Lack of fit: Not significant; R-Squared: 0.997.

**Table 4 ijms-24-09327-t004:** Central Composite Design (CCD) matrix and experimentally observed responses in Experimental Setup 2.

Exp No	Independent Variables		Observed Response						
			Encapsulated Phenolic Compound Concentration (mg/mL)						
Std	Silk Fibroin Concentration (mg/mL)	EtOH Concentration (%)	Particle Size (nm)	Trans- Resveratrol	Quercetin	Gallic Acid	Vanillic Acid	Total Phenol Amount	Phenol mg/mg Silk
1	1	60	150	1.79	1.22	0	0	3	0
2	2	60	50	1.58	1.22	0.23	2.2	5.22	2
3	1	80	190	1.49	0.615	0	1.9	4.01	2.67
4	2	80	100	1.82	1.012	0.13	2	4.96	1.66
5	0.9	70	130	1.54	0.92	0	0	2.45	1.82
6	2.1	70	150	2.13	1.22	0.23	0	3.57	1.13
7	1.5	58	250	1.83	2.12	3 × 10^−2^	0	3.97	1.77
8	1.5	82	150	1.63	0	3 × 10^−2^	2.2	3.86	1.72
9	1.5	70	200	1.91	1.02	0	3.1	6.03	2.68
10	1.5	70	203	1.94	1.22	0	2.8	5.95	2.65
11	1.5	70	201	1.92	1.82	0	2.9	6.64	2.95
12	1.5	70	204	1.93	1.12	0	2.9	5.95	2.64

**Table 5 ijms-24-09327-t005:** Results of ANOVA for encapsulated vanillic acid concentration, trans-resveratrol concentration, quercetin concentration, gallic acid concentration, and particle size by polynomial second-order models.

Response	Source	SS	DF	MS	F-Value	*p*-Value	R^2^
Particle Size	Model	32,675.67	8	4084.46	1225.34	<0.0001	0.99
	A-SF Concentration	200	1	200	60	0.0045	
	B-EtOH Concentration	5000	1	5000	1500	<0.0001	
	Pure Error	10	3	3.33			
	Total	32,685.67	11				
Vanillic Acid Concentration	Model	0.000	8	2.2 × 10^−6^	143.52	0.0009	0.99
	A-SF Concentration	0.000	1	0.0000	2.84	1.0000	
	B-EtOH Concentration	2.4 × 10^−6^	1	2.4 × 10^−6^	432.32	0.0011	
	Pure Error	4.7 × 10^−8^	3	1.5 × 10^−8^			
	Total	0	11				
Trans-resveratrol Concentration	Model	4.1 × 10^−7^	8	5.1 × 10^−8^	309.84	0.0003	0.99
	A-SF Concentration	1.7 × 10^−7^	1	1.7 × 10^−7^	1044.30	<0.0001	
	B-EtOH Concentration	2.0 × 10^−8^	1	2.0 × 10^−8^	120	0.0016	
	Pure Error	5.0 × 10^−10^	3	1.6 × 10^−10^			
	Total	4.1 × 10^−7^	11				
Quercetin Concentration	Model	2.566 × 10^−6^	8	3.2 × 10^−7^	21.07	0.015	0.98
	A-SF Concentration	4.5 × 10^−8^	1	4.5 × 10^−8^	0.35	0.18	
	B-EtOH Concentration	2.2 × 10^−6^	1	2.2 × 10^−6^	85.51	0.001	
	Pure Error	4.5 × 10^−8^	3	1.5 × 10^−8^			
	Total	2.6 × 10^−6^	11				
Gallic Acid Concentration	Model	8.8 × 10^−8^	5	1.7 × 10^−8^	84.38	<0.0001	0.98
	A-SF Concentration	5.8 × 10^−8^	1	5.9 × 10^−8^	281.75	<0.0001	
	B-EtOH Concentration	1.4 × 10^−9^	1	1.4 × 10^−9^	6.97	0.0386	
	Pure Error	0	3	0			
	Total	8.9 × 10^−8^	11				

SS: Sum of Squares; DF: degree of freedom; MS: mean square.

**Table 6 ijms-24-09327-t006:** Central Composite Design (CCD) matrix and experimentally observed responses in Experimental Setup 3.

Exp No	Independent Variables			Observed Response						
					Encapsulated Phenolic Compound Concentration					
Std	Silk Fibroin Concentration (mg/mL)	EtOH Concentration (%)	pH	Particle Size (nm)	Gallic Acid (µg/mL)	Vanillic Acid (µg/mL)	Trans-Resveratrol (µg/mL)	Quercetin (µg/mL)	Total Phenol Amount (µg/mL)	Phenol mg/mg × 10^−3^ Silk
1	1	60	2	250	0.8	0.3	1.7	1.01	3.81	3.81
2	2	60	2	303	0.3	0.3	1.7	0.91	3.21	1.07
3	1	80	2	402	0	0	2.1	0.91	3.01	2.01
4	2	80	2	302	0	0	1.4	0.71	2.11	0.70
5	1	60	7	120	0.2	0	2.4	1.21	3.81	2.54
6	2	60	7	120	0	0	2.3	1.01	3.31	1.10
7	1	80	7	380	0.1	0	1.6	1.01	2.71	1.81
8	2	80	7	303	0.4	0	2.4	1.21	4.01	1.34
9	0.9	70	4.5	302	0.2	0	1.6	0.71	2.51	1.86
10	2.1	70	4.5	500	0.1	0.2	1.9	0.81	3.01	0.96
11	1.5	58	4.5	550	0	0	1.7	0	1.7	0.76
12	1.5	82	4.5	303	0	0	1.1	0	1.1	0.49
13	1.5	70	1.5	500	0.2	0	1.8	1.11	3.11	1.38
14	1.5	70	7.5	500	0.2	0	1.2	0	1.4	0.62
15	1.5	70	4.5	403	0	0	1.6	0.7	2.3	1.02
16	1.5	70	4.5	405	0	0	1.6	0.7	2.3	1.02
17	1.5	70	4.5	402	0.1	0	1.7	0.91	2.71	1.20
18	1.5	70	4.5	377	0.1	0.1	1.7	0.91	2.81	1.25

**Table 7 ijms-24-09327-t007:** Results of ANOVA for optimized encapsulated vanillic acid concentration, trans-resveratrol concentration, quercetin concentration, gallic acid concentration, and particle size by polynomial second-order models.

Response	Source	SS	DF	MS	F-Value	*p*-Value	R^2^
Particle Size	Model	2.4 × 10^5^	14	17,575.31	100.48	0.0014	0.99
	A-SF Concentration	19,602.00	1	19,602.00	112.06	0.0018	
	B-EtOH Concentration	30,504.50	1	30,504.50	174.39	0.0009	
	pH	0	1	0	0	0	
	Pure Error	524.75	3	174.92			
	Total	2.466 × 10^5^	17				
Vanillic Acid Concentration	Model	1.775 × 10^−7^	11	2.2 × 10^−6^	12.91	0.0026	0.95
	A-SF Concentration	2.000 × 10^−8^	1	2.0 × 10^−8^	16.00	0.0071	
	B-EtOH Concentration	0.0000	1	0.0000	0.0000	1.000	
	pH	0.0000	1	0.0000	0.0000	1.000	
	Pure Error	7.500 × 10^−9^	3	2.5 × 10^−9^			
	Total	1.850 × 10^−7^	17				
Trans-resveratrol Concentration	Model	2.2 × 10^−6^	14	1.5 × 10^−7^	47.89	0.0043	0.99
	A-SF Concentration	4.5 × 10^−8^	1	4.5 × 10^−8^	13.50	0.0349	
	B-EtOH Concentration	1.8 × 10^−7^	1	1.8 × 10^−7^	54.00	0.0052	
	pH	1.8 × 10^−7^	1	1.8 × 10^−7^	1.50	0.0052	
	Pure Error	1.0 × 10^−8^	3	3.3 × 10^−9^			
	Total	2.2 × 10^0^	17				
Quercetin Concentration	Model	2.5 × 10^−6^	14	1.7 × 10^−7^	12.18	0.0313	0.98
	A-SF Concentration	5.0 × 10^−9^	1	5.0 × 10^−9^	0.3401	0.6007	
	B-EtOH Concentration	0.0000	1	0.0000	0.0000	1.000	
	pH	6.1 × 10^−7^	1	6.1 × 10−^7^	41.91	0.0075	
	Pure Error	4.4 × 10^−8^	3	1.4 × 10^−8^			
	Total	2.5 × 10^−6^	17				
Gallic Acid Concentration	Model	6.4 × 10^−7^	9	7.2 × 10^−8^	15.24	0.0004	0.94
	A-SF Concentration	2.4 × 10^−8^	1	2.5 × 10^−8^	5.27	0.0509	
	B-EtOH Concentration	5.8 × 10^−8^	1	5.9 × 10^−8^	12.46	0.0077	
	pH	1.4 × 10^−8^	1	1.5 × 10^−8^	3.12	0.1155	
	Pure Error	1.0 × 10^−8^	3	3.3 × 10^−9^			
	Total	6.8 × 10^7^	17				

**Table 8 ijms-24-09327-t008:** Separate Encapsulation of Phenolic Compounds.

Phenol Compound	Initial Concentration (µg/mL)	Final Concentration (µg/mL)	Encapsulated Phenolic Compound (µg/mL)	Phenol (µg/mg 10^−3^) Silk Fibroin
Gallic Acid	5.82	5.80	0	0
Vanillic Acid	3.4	3.4	0	0
Trans-resveratrol	07.03	5.1	1.93	0.65
Quercetin	10.3	10.0	1.31	0.44

**Table 9 ijms-24-09327-t009:** TLC screening parameters.

Sample\Rf Values	Unloaded Phenols	Loaded Phenols
PFmix	0.3 and 0.19	0.32 and 0.19
Trans-Resveratrol	0.25	0.2
Gallic Acid	0.057	0.06
Vanillic Acid	0.34	0.32
Quercetin	0.23	0.13
Control	-	-

**Table 10 ijms-24-09327-t010:** Independent variables and their levels used for CCD.

Experimental Setup	Independent Variables	Unit	Factors’ Symbol	Coded Levels	
				−1	1
Set 1	Silk Fibroin Concentration	mg/ml	A	1	2
Set 2	Ethanol concentration in Solution	%	B	60	80
Set 3	pH		C	2	7

## Data Availability

Not applicable.

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
