# Peer review of "Selective Encapsulation of the Polyphenols on Silk Fibroin Nanoparticles: Optimization Approaches"

_ijms, 2023, doi:10.3390/ijms24119327_

Round 1
Reviewer 1 Report
In this article, the authors proposed a statistical method to define better parameters to obtain silk fibroin nanoparticles loaded with polyphenols.
I suggested some modifications.
Comments:
- In my opinion, the title is not appropriate for this manuscript. The actual title does not describe the research.
- Introduction is too long. Please reduce the introduction length.
- The cut-off of the dialysis membrane used for silk fibroin solution is not indicated (page 21).
- Preparation of silk fibroin nanoparticles: the “various concentration of silk fibroin solutions” is not specified. Please better described all methods. I suggest adding a table in the materials and method section with all variables considered.
- Figure 1: please indicate in the figure the shifted peaks/bands.
- Figure 7 presented a low resolution.
- Is not clear the method used to solubilize the nanoparticles and for the determination of drug loading.
Minor editing of English language required
Reviewer 2 Report
The authors describe in this manuscript the optimization of the encapsulation of phenolic compounds in silk fibroin nanoparticles. The study is completed with the characterization of these nanoparticles evaluating morphology, efficiency of the encapsulation, and antioxidant properties. The experiments have been systematically performed and the manuscript is well organized.
I suggest that this manuscript can be published in this journal after the completion of the following issues:
- the information about the physical stability of these nanoparticles.
- the authors should compare the results obtained for the antioxidant activity of nanoparticles with the activity of pure polyphenols and with other data from the literature
- there are some typing errors in the text, please correct them - lines 53, 93, 118, 505 (K2S2O8), 539, 548, 554,
The topic of the manuscript is relevant to the publication policy of the Int. J. Mol. Sci. and, based on the above comments, I would suggest publishing the manuscript after minor revision.
In the manuscript, there are some typing errors.
Round 2
Reviewer 1 Report
Authors modified the manuscript as requested.